# Reporting of Acute Inflammatory Neuropathies with COVID-19 Vaccines: Subgroup Disproportionality Analyses in VigiBase

**DOI:** 10.3390/vaccines9091022

**Published:** 2021-09-14

**Authors:** Roberta Noseda, Paolo Ripellino, Sara Ghidossi, Raffaela Bertoli, Alessandro Ceschi

**Affiliations:** 1Division of Clinical Pharmacology and Toxicology, Institute of Pharmacological Sciences of Southern Switzerland, Ente Ospedaliero Cantonale, 6900 Lugano, Switzerland; Roberta.noseda@eoc.ch (R.N.); Sara.ghidossi@eoc.ch (S.G.); raffaela.bertoli@eoc.ch (R.B.); 2Neurology Department, Neurocenter of Southern Switzerland, Ente Ospedaliero Cantonale, 6900 Lugano, Switzerland; Paolo.ripellino@eoc.ch; 3Clinical Trial Unit, Ente Ospedaliero Cantonale, 6900 Lugano, Switzerland; 4Faculty of Biomedical Sciences, Università della Svizzera Italiana, 6900 Lugano, Switzerland; 5Department of Clinical Pharmacology and Toxicology, University Hospital Zurich, 8091 Zurich, Switzerland

**Keywords:** acute inflammatory neuropathy, COVID-19 vaccines, VigiBase, safety, disproportionality

## Abstract

Since marketing authorization, cases of neuralgic amyotrophy (NA), facial paralysis/Bell’s palsy (FP/BP), and Guillain-Barré syndrome (GBS) were reported with COVID-19 vaccines of different technologies. This study aimed to assess whether NA, FP/BP, and GBS were more frequently reported in VigiBase with COVID-19 vaccines (of any technologies) than with other viral vaccines, over the full database and across potential risk groups by sex and age. The reporting odds ratio (ROR) with 95% confidence interval (95% CI) was used as the measure of disproportionality and subgroup disproportionality analyses were performed by sex and age. Out of 808,906 safety reports with COVID-19 vaccines, 57 (0.01%) reported NA, 3320 (0.4%) FP/BP, and 632 (0.1%) GBS. There were not signals of disproportionate reporting for NA and GBS with COVID-19 vaccines against other viral vaccines. FP/BP was disproportionately more frequently reported with COVID-19 vaccines than with other viral vaccines over the full database (ROR 1.12, 95%CI 1.07–1.17), in males (ROR 1.65, 95%CI 1.54–1.78) and in age subgroups 65–74 years (ROR 1.21, 95%CI 1.05–1.39) and ≥75 years (ROR 1.84, 95%CI 1.52–2.22). Albeit not proving causation, these findings might support clinicians in decision-making for patients potentially at risk for developing an acute inflammatory neuropathy with COVID-19 vaccines.

## 1. Introduction

Neuralgic amyotrophy (NA), facial paralysis/Bell’s palsy (FP/BP), and Guillain-Barré syndrome (GBS) are acute inflammatory neuropathies whose potential pathophysiological triggers include antecedent infections, traumatic injuries, malignancies, and autoimmune disorders [1,2,3]. Notably, based on biological plausibility and temporal association, the onset of these acute inflammatory neuropathies was suggested to also be linked with vaccinations, most often the influenza vaccine [1,2,3].

Since marketing authorization, cases of NA, FP/BP and GBS were reported with COVID-19 vaccines [4,5,6,7,8,9,10,11]. A recent pharmacovigilance study in VigiBase, the World Health Organization’s (WHO) global database of suspected adverse drug reactions (ADRs) found that the reporting frequency of FP/BP with mRNA COVID-19 vaccines was similar to that with other viral vaccines [12]. 

In pharmacovigilance, disproportionality analysis by subgroups represents a useful methodological approach to uncover risk groups for ADRs [13]. Notably, in large international spontaneous databases, such as VigiBase, subgroup disproportionality analysis performs better than crude analysis (in the full database) and stratified analysis, in both sensitivity and precision [14].

The present study performed in VigiBase aimed to assess whether NA, FP/BP, and GBS were disproportionately more frequently reported with COVID-19 vaccines (of any technologies) than with other viral vaccines (overall and restricted to influenza vaccines), over the full database and across potential risk subgroups by sex and age.

## 2. Materials and Methods

De-duplicated safety reports of COVID-19 vaccine-related NA, FP/BP, and GBS, collected in VigiBase as of 16 May 2021, were retrieved. COVID-19 vaccines were selected based on the WHO drug standardized drug grouping “Vaccines for COVID-19” that includes COVID-19 vaccines of any technologies; NA, FP/BP, and GBS ADRs were selected based on the correspondent preferred terms according to the Medical Dictionary for Regulatory Activities (MedDRA, version 24.0, https://www.meddra.org/ accessed on 13 September 2021).

The baseline characteristics of COVID-19 vaccine-related safety reports of NA, FP/BP, and GBS included patient sex and age, reporter qualification, suspected COVID-19 vaccine, time-to-onset, and outcome of the acute inflammatory neuropathy of interest.

In disproportionality analyses, either safety reports concerning NA, FP/BP, and GBS, or other ADRs, reported with any other viral vaccines (of the anatomical therapeutic chemical group, ATC, J07B-excluding COVID-19 vaccines), were used as comparators groups. The reporting odds ratio (ROR) with the 95% confidence interval (95% CI) was used as measure of disproportionality and computed when ≥5 safety reports with the acute inflammatory neuropathy of interest were present. Signals of disproportionate reporting were characterized by 95% CI lower bound >1. 

Subgroup disproportionality analyses were performed by sex and age (<18 years, 18–44 years, 45–64 years, 65–74 years, ≥75 years). Safety reports concerning NA, FP/BP, and GBS, or other ADRs, reported with influenza vaccines (ATC J07BB) were used as comparator groups in sensitivity disproportionality analyses performed both on the full database and across risk subgroups. In subgroup disproportionality analyses, RORs were calculated within each group separately and a signal was counted if the score from any subgroups met signal criteria. Safety reports with missing values for the two variables used to define subgroups were excluded from the analyses.

Data management and analysis were carried out by Microsoft Excel (2010, Microsoft Corporation, Washington, USA) and GraphPad Prism 9 (GraphPad Software Inc., San Diego, California, USA).

According to the Swiss Human Research Act 810.30 (status as of 26 May 2021), ethical approval by the local Ethical Committee was not required for this study as it involved anonymized health-related data.

## 3. Results

Up to 16 May 2021, 808,906 COVID-19 vaccine-related safety reports have reached VigiBase. Of these, 57 (0.01%) reported NA, 3320 (0.4%) FP/BP, and 632 (0.1%) GBS. Table 1 shows the baseline characteristics of COVID-19 vaccine-related safety reports concerning NA, FP/BP, and GBS. COVID-19 vaccine-related NA and FP/BP more frequently involved female patients (n = 40, 70.2% and n = 1994, 60.1%, respectively), whereas GBS more frequently involved males (n = 345, 54.6%). The median age of the patients reporting COVID-19 vaccine-related NA was 49 years (interquartile range, IQR, 41-57 years, n = 54), 54 (IQR 42–68 years, n = 3021) for patients reporting COVID-19 vaccine-related FP/BP, and 60 (IQR 49–68 years, n = 581) for patients reporting COVID-19 vaccine-related GBS. NA and GBS were more frequently reported by physicians (n = 21, 36.8% and n = 269, 42.6%, respectively), whilst FP/BP was more frequently reported by consumers/not health professionals (n = 773, 23.3%). The Pfizer BioNTech COVID-19 vaccine was the most frequently suspected COVID-19 vaccine for NA (n = 38, 66.7%) and FP/BP (n = 1822, 54.9%), whereas the COVID-19 Vaccine AstraZeneca for GBS (n = 327, 51.7%) (Table 1).

In the disproportionality analyses performed for NA, overall, no signals of disproportionate reporting with COVID-19 vaccines were detected against either other viral vaccines (ROR 0.23, 95% CI 0.17–0.30) or influenza vaccines (ROR 0.12, 95% CI 0.09–0.16). Neither sex nor age emerged as risk factors for disproportionate reporting of NA with COVID-19 vaccines, compared either to other viral vaccines or to influenza vaccines (Table 2).

Additionally, for GBS, no signals of disproportionate reporting with COVID-19 vaccines were found, neither overall (against other viral vaccines, ROR 0.15, 95% CI 0.13–0.16; against influenza vaccines, ROR 0.06, 95% CI 0.05–0.06) nor by sex and age subgroup disproportionality analyses (Table 2). 

In the overall analysis on the full database, FP/BP was disproportionately more frequently reported with COVID-19 vaccines than with other viral vaccines (ROR 1.12, 95% CI 1.07–1.17). Such a signal of disproportionate reporting was not detected in the sensitivity disproportionality analysis against influenza vaccines (ROR 0.75, 95% CI 0.71–0.80). Subgroup disproportionality analyses showed signals of disproportionate reporting for FP/BP with COVID-19 vaccines against other viral vaccines in males (ROR 1.65, 95% CI 1.54–1.78) and in individuals of advanced age (in the age subgroup 65-74 years, ROR 1.21, 95% CI 1.05–1.39; in the age subgroup ≥75 years, ROR 1.84, 95% CI 1.52–2.22). Sensitivity analyses against influenza vaccines confirmed signals of disproportionate reporting for FP/BP in the age subgroup 65–74 years (ROR 1.46, 95% CI 1.22–1.75) and in the age subgroup ≥75 years (ROR 1.76, 95% CI 1.41–2.20) (Table 2).

## 4. Discussion

By querying VigiBase, this study found that the reporting frequency of NA and GBS with COVID-19 vaccines of different technologies was similar to that of other viral vaccines, and neither patient sex nor age were risk factors. Conversely, a signal of disproportionate reporting was observed for FP/BP with COVID-19 vaccines in the analysis on the full database. Subgroup disproportionality analyses highlighted male sex and advanced patient age (≥65 years) as risk factors for the reporting of FP/BP with COVID-19 vaccines against other viral vaccines. Notably, the safety signals for COVID-19 vaccine-related FP/BP detected in advanced age subgroups were confirmed in the sensitivity analysis against influenza vaccines.

The reporting frequency of FP/BP with COVID-19 vaccines in VigiBase was uncommon, those of NA and GBS were rare, reflecting the background incidence rates of these three acute inflammatory neuropathies [1,2,3]. Notably, most of the safety reports of NA, FP/BP and GBS were reported with the Pfizer BioNTech COVID-19 vaccine, the COVID-19 Vaccine Moderna, and the COVID-19 Vaccine AstraZeneca, whilst less frequently were associated with the Janssen COVID-19 vaccine, for which, as for the COVID-19 Vaccine AstraZeneca, rather an association with unusual blood clot formation was reported [15].

Because of the new technology, the use by a large number of individuals in a brief period, and the recommendation of an additional dose at least for immunocompromised patients, concerns about the safety of mRNA COVID-19 vaccines have emerged. Cases of FP/BP were reported with mRNA COVID-19 vaccines but have also occurred with COVID-19 vaccines of different technologies [6,7,8]. Indeed, the biological mechanism linking FP/BP onset and COVID-19 vaccines is likely related to immune activation and independent on vaccine technology [16].

Therefore, by broadening the spectrum of safety reports of interest to those with COVID-19 vaccines (of all technologies) suspected of being associated with FP/BP onset, the present study controverts the one previously performed in VigiBase by Renoud et al., which found that the reporting rate of FP/BP after mRNA COVID-19 vaccines was not higher than that observed with other viral vaccines [12]. In that study, authors used stratification to control confounding factors (including sex and age) and, assuming the absence of risk variation across strata, provided a single combined (not significant) disproportionality measure [12]. Nevertheless, ignoring the diversity within a dataset may result in signals being masked. When a confounding factor acts as an effect modifier, subgroup disproportionality analysis can highlight risk variation across subgroups by computing a disproportionately measure within each subgroup [13]. Consistently, the present study uncovered that reporting of FP/BP with COVID-19 vaccines was disproportionately more frequent in males and in patients aged ≥65 years. 

As passive surveillance system, VigiBase suffers of underreporting and selective reporting, lack of clinical details, missing and/or inaccurate data. Safety reports reflect suspicions without being necessarily indicative of a causal relationship between the suspected drug/vaccine and the reported adverse event. Moreover, lacking information on the number of patients exposed to a certain drug/vaccine, VigiBase does not allow defining the incidence of adverse events.

## 5. Conclusions

Subgroup disproportionality analyses in VigiBase on safety reports of NA, FP/BP, and GBS with COVID-19 vaccines did not detect any signals for NA and GBS, whereas highlighted that FP/BP was disproportionately more frequently reported with COVID-19 vaccines than with other viral vaccines. Moreover, male sex and advanced age (≥65 years) were risk factors for increased reporting of FP/BP with COVID-19 vaccines.

Awareness of these findings might support clinicians in decision making when confronted with patients who are potentially at risk for developing an acute inflammatory neuropathy following COVID-19 vaccine administration.

## Figures and Tables

**Table 1 vaccines-09-01022-t001:** Baseline characteristics of COVID-19 vaccine-related safety reports of neuralgic amyotrophy, facial paralysis/Bell’s palsy, and Guillain-Barré syndrome, collected in VigiBase as of 16 May 2021.

Characteristic	Neuralgic Amyotrophy (n = 57)	Facial Paralysis/Bell’s Palsy (n = 3320)	Guillain-Barré Syndrome (n = 632)
**Sex**			
Female	40 (70.2)	1994 (60.1)	282 (44.6)
Male	17 (29.8)	1280 (38.6)	345 (54.6)
Not reported	-	46 (1.4)	5 (0.8)
**Age, years**			
<18	-	8 (0.2)	1 (0.2)
18–44	22 (38.6)	891 (26.8)	112 (17.7)
45–64	24 (42.1)	1197 (36.1)	272 (43.0)
65–74	6 (10.5)	479 (14.4)	111 (17.6)
≥75	2 (3.5)	446 (13.4)	85 (13.4)
Not Reported	3 (5.3)	299 (9.0)	51 (8.1)
**Reporter Qualification**			
Physician	21 (36.8)	636 (19.2)	269 (42.6)
Pharmacist	1 (1.8)	89 (2.7)	34 (5.4)
Other Health Professional	2 (3.5)	205 (6.2)	38 (6.0)
Consumer/Non Health	8 (14.0)	773 (23.3)	84 (13.3)
Professional			
Not Reported	25 (43.9)	1641 (49.4)	215 (34.0)
**Suspected COVID-19 Vaccine**			
**(WHO Drug Trade Name)**			
Pfizer BioNTech COVID-19	38 (66.7)	1822 (54.9)	164 (25.9)
Vaccine			
COVID-19 Vaccine Moderna	14 (24.6)	623 (18.8)	75 (11.9)
COVID-19 Vaccine AstraZeneca	5 (8.8)	671 (20.2)	327 (51.7)
Janssen COVID-19 Vaccine	-	162 (4.9)	58 (9.2)
Unknown (Covid-19 Vaccine)	-	27 (0.8)	4 (0.6)
CoronaVac	-	13 (0.4)	3 (0.5)
Convidecia	-	1 (0.0)	-
Sputnik V	-	1 (0.0)	1 (0.2)
**Time-to-onset**			
With respect to one dose *	48 (84.2)	2767 (83.3)	514 (81.3)
Median (IQR), days	4 (1–13)	4 (1–12)	11 (5–17)
With respect to the 2nd dose	3 (5.3)	49 (1.5)	7 (1.1)
Median (IQR), days	8 (5–13)	10 (1–22)	10 (5–12)
Not reported	6 (10.5)	504 (15.2)	111 (17.6)
**Outcome**			
Recovered	1 (1.8)	544 (16.4)	29 (4.6)
Recovered with Sequelae	-	20 (0.6)	8 (1.3)
Recovering	8 (14.0)	423 (12.7)	149 (23.6)
Not Recovered	21 (36.8)	647 (19.5)	186 (29.4)
Death	-	2 (0.0)	8 (1.3)
Not Reported	27 (47.4)	1684 (50.7)	252 (39.9)

Data are No. (%); * The reporter recorded one dose but it is unknown whether it was the first or the second dose; Abbreviations: IQR, interquartile range.

**Table 2 vaccines-09-01022-t002:** Disproportionality analysis of COVID-19 vaccine-related safety reports of neuralgic amyotrophy, facial paralysis/Bell’s palsy, and Guillain-Barré syndrome versus all other viral vaccines and influenza vaccines (dataset date 16 May 2021).

	COVID-19 Vaccines		Other Viral Vaccines			Influenza Vaccines		
	No. of Cases	Total No.	No. of Cases	Total No.	ROR (95% CI)	No. of Cases	Total No.	ROR (95% CI)
**Neuralgic Amyotrophy**								
Overall	57	780,073	339	1,057,529	0.23 (0.17–0.30)	166	263,340	0.12 (0.09–0.16)
By sex subgroup								
Males	17	199,520	172	408,786	0.20 (0.12–0.33)	88	87,506	0.08 (0.05–0.14)
Females	40	566,179	164	625,197	0.27 (0.19–0.38)	77	171,560	0.16 (0.11–0.23)
By age subgroups								
<18 years	0	3449	27	493,510	NA	6	49,908	NA
18–44 years	22	303,585	96	191,690	0.14 (0.09–0.23)	37	60,179	0.12 (0.07–0.20)
45–64 years	24	278,936	120	134,691	0.10 (0.06–0.15)	61	65,187	0.09 (0.06–0.15)
65–74 years	6	77,812	45	68,080	0.12 (0.05–0.27)	34	38,370	0.09 (0.04–0.21)
≥75 years	2	61,711	14	37,007	NC	8	23,587	NC
**Facial Paralysis/Bell’s Palsy**								
Overall	3320	780,073	4031	1,057,529	1.12 (1.07–1.17)	1492	263,340	0.75 (0.71–0.80)
By sex subgroup								
Males	1280	199,520	1589	408,786	1.65 (1.54–1.78)	643	87,506	0.87 (0.79–0.96)
Females	1994	566,179	2428	625,197	0.91 (0.85–0.96)	846	171,560	0.71 (0.66–0.77)
By age subgroups								
<18 years	8	3449	1105	493,510	1.04 (0.52–2.08)	201	49,908	0.57 (0.28–1.17)
18–44 years	891	303,585	1106	191,690	0.51 (0.46–0.55)	448	60,179	0.39 (0.35–0.44)
45–64 years	1197	278,936	893	134,691	0.65 (0.59–0.70)	498	65,187	0.56 (0.50–0.62)
65–74 years	479	77,812	347	68,080	1.21 (1.05–1.39)	162	38,370	1.46 (1.22–1.75)
≥75 years	446	61,711	146	37,007	1.84 (1.52–2.22)	97	23,587	1.76 (1.41–2.20)
**Guillan-Barré syndrome**								
Overall	632	780,073	5832	1,057,529	0.15 (0.13–0.16)	3613	263,340	0.06 (0.05–0.06)
By sex subgroup								
Males	345	199,520	2990	408,786	0.24 (0.21–0.26)	1987	87,506	0.07 (0.07–0.08)
Females	282	566,179	2749	625,197	0.11 (0.10–0.13)	1570	171,560	0.05 (0.05–0.06)
By age subgroups								
<18 years	1	3449	926	493,510	NC	220	49,908	NC
18–44 years	112	303,585	1209	191,690	0.06 (0.05–0.07)	647	60,179	0.03 (0.03–0.04)
45–64 years	272	278,936	1587	134,691	0.08 (0.07–0.09)	1185	65,187	0.05 (0.05–0.06)
65–74 years	111	77,812	881	68,080	0.11 (0.09–0.13)	723	38,370	0.07 (0.06–0.09)
≥75 years	85	61,711	610	37,007	0.08 (0.07–0.10)	523	23,587	0.06 (0.05–0.08)

Abbreviations: ROR, reporting odds ratio; CI, confidence interval, NA, not applicable; NC, not calculated (less than five safety reports).

## Data Availability

The data presented in this study are available from VigiBase upon formal request to the Uppsala Monitoring Centre-WHO Collaborating Centre for International Drug Monitoring.

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
