# Peer review of "Reporting of Acute Inflammatory Neuropathies with COVID-19 Vaccines: Subgroup Disproportionality Analyses in VigiBase"

_vaccines, 2021, doi:10.3390/vaccines9091022_

Round 1
Reviewer 1 Report
In this paper, the authors analysed the number of cases of neuralgic amyotrophy (NA), facial paralysis/Bell’s palsy (FP/BP), and Guillain-Barrè syndrome (GBS), reported with COVID-19 vaccines in VigiBase® .
They considered vaccines of different technologies and observed thatthe reporting frequency of NA and GBS was similar to that with other viral vaccines. Conversely, male sex and advanced patient age (≥65 years) were highlighted as risk factors for FP/BP with COVID-19 vaccines against other viral vaccines.
The analysis is well done and the results is clearly evidenced by Table 1 and Table 2. My only concern is that the data reported in Table 2 are not adequately described in the text and their analysis can be improved.
Reviewer 2 Report
The Covid-19 vaccines that have been in use for several months now, though remarkably effective, are not without controversy. Some of the reluctance to take these vaccines is largely based in general vaccine skepticism. However, these so-called “anti-vaxxers” have seized upon the extremely low percentage of complications that have been associated with Covid vaccines. These complications include neuropathic manifestations such as the relatively rare Guillain-Barre syndrome (GBS), Bell’s palsy and other instances of facial paralysis (FP/BP), as well as neuralgic amyotrophy (NA). However, such complications have also been associated with other vaccines including influenza. Indeed, such associations are major factors in the genesis of the strident opposition to acceptance of vaccines of any kind. This skepticism was even further exacerbated by misunderstanding about the speed with which the Covid vaccines were developed. A fairly large segment of the public feel that these vaccines were “rushed” into Emergency Use Authorization without proper vetting of efficacy and safety.
Thus, it is important that we obtain hard data concerning the prevalence of each of these syndromes in patients who received one of the Covid vaccines and compare it to their general prevalence in the overall population and in patients receiving other vaccines. This is especially critical for the mRNA-based vaccines (Pfizer and Moderna), which are the first two such vaccines administered to such a high percentage of the population. Indeed, the Johnson & Johnson Covid vaccine has actually been the most scrutinized due its association with blood clots not to mention its single shot regimen.
In this manuscript, a highly detailed analysis of the frequency of each of the above neurological syndromes is performed in more than 800,000 immunizations of one of the Covid-19 vaccines. The goal is to determine whether any of these syndromes is disproportionately associated with Covid vaccination and, if so, whether it is more prevalent in one sex or the other or in a particular age group. There is no such disproportionality associated with NA (only 0.01%). Though there are cases of GBS in patients receiving the Covid vaccines, it is far more prevalent with other vaccines, including influenza. However, facial paralysis and Bell’s palsy are quite different in this regard. The prevalence of these syndromes is 0.4% of patients receiving a Covid vaccine, which is disproportionately higher than with other vaccines. Moreover, males over the age of 65 show an even more disproportionate prevalence of these syndromes.
An important caveat in the disproportionate findings with FP/BP, but not NA or GBS, is that the former are largely self-reported, while the latter two are reported by health professionals. So, one may question the accuracy of the data concerning FP/BP. To their credit, the authors do acknowledge this caveat. Based on this analysis, it is prudent for clinicians to counsel patients suffering from these syndromes about the risks involved in receiving one of the Covid vaccines.
This is considered a very well-performed analysis that raises our awareness of a possible causative relationship between the Covid vaccines and FP/BP and alerts the medical profession to be cautious in vulnerable patients. However, there are a few minor points, which, if addressed, could improve the manuscript. As mentioned above, the two mRNA vaccines are very new and widely used. This wide use of such a new technology has obviously been fodder to those reluctant to be vaccinated. I feel the authors should acknowledge this issue somewhere in the manuscript. In addition, while the concern over the development of blood clots associated with the J&J Covid vaccine is obviously not a part of this study, the authors should at least acknowledge this issue, as it is perhaps the best-known potential complication from these vaccines. A third point that it would behoove the authors to at least mention is that a third dose of the mRNA vaccines is called for. Indeed, this third shot is already being administered to immunocompromised individuals. But, these points do not detract from the overall completeness of the study.
